# Energy-Level Jumping Algorithm for Global Optimization in Compressive Sensing-Based Target Localization

**DOI:** 10.3390/s19112502

**Published:** 2019-05-31

**Authors:** Tianjing Wang, Xinjie Guan, Xili Wan, Guoqing Liu, Hang Shen

**Affiliations:** 1School of Computer Science and Technology, Nanjing Tech University, Nanjing 211816, China; xgck9@mail.umkc.edu (X.G.); xiliwan@njtech.edu.cn (X.W.); hshen@njtech.edu.cn (H.S.); 2School of Physical and Mathematical Sciences, Nanjing Tech University, Nanjing 211816, China; guoqing@njtech.edu.cn

**Keywords:** target localization, compressive sensing, global optimization, locally optimal sparse solution, globally optimal sparse solution, energy-level jumping

## Abstract

Target localization is one of the essential tasks in almost applications of wireless sensor networks. Some traditional compressed sensing (CS)-based target localization methods may achieve low-precision target localization because of using locally optimal sparse solutions. Solving global optimization for the sparse recovery problem remains a challenge in CS-based target localization. In this paper, we propose a novel energy-level jumping algorithm to address this problem, which achieves high-precision target localization by solving the globally optimal sparse solution of $l_{p}$-norm (0<p<1) minimization. By repeating the process of energy-level jumping, our proposed algorithm establishes a global convergence path from an initial point to the global minimizer. Compared with existing CS-based target localization methods, the simulation results show that our localization algorithm obtain more accurate locations of targets with the significantly reduced number of measurements.

## 1. Introduction

Driven by the commercial and military applications, localization for people, vehicles and devices has attracted increasing attention in Internet of Things [1,2,3]. The traditional satellite-based target localization system can be exploited in favorable outdoor scenarios, but this system cannot be widely used in indoor scenarios because of the lack of the sight to satellite and the high cost of the equipped wireless devices. The wide deployment of wireless sensor networks (WSNs) provides an attractive choice for target localization, and much work has been done to develop target localization and tracking methods by applying various types of WSNs [4,5].

Without the need of any additional hardware for sensors, the early received signal strength (RSS)-based localization system [6,7] located targets by measuring the values of RSS from targets. Combining with fingerprinting, the RSS-fingerprinting method creates a signature map to represent the nature of radio propagation in physical space. A typical algorithm called *k*-nearest neighbor algorithm [8] obtains the locations of unknown targets by comparing the newly detected RSS measurements with the signature map. However, such system has to exchange a large number of data between sensors and a fusion center that greatly consumes the limited computation resource and network bandwidth. It becomes a challenging task to reduce the number of the RSS measurements for saving network resource while maintaining high positioning accuracy.

To deal with this challenge, compressive sensing (CS) [9] has been introduced to locate targets from the less number of measurements in terms of the intrinsic sparse nature of target localization [10]. Formally, the CS-based target localization problem can be transformed into the sparse recovery of $l_{1}$-norm minimization [11,12]. A sparse solution modelling the locations of targets was then reconstructed by the basis pursuit (BP) algorithm [13] with high computational complexity. To reduce computational cost, the orthogonal matching pursuit (OMP) algorithm [14] in a greedy manner was applied to estimate the locations of targets by $l_{0}$-norm minimization. Another greedy matching pursuit (GMP) algorithm [15] tried to select an optimal 1-sparse vector at each iteration (i.e., the location of a target), which inevitably led to much localization time. To achieve high position accuracy, these greedy algorithms need the larger number of measurements that greatly increases the sensing burden and energy consumption of sensors. Therefore, it is necessary to find more efficient method to reconstruct a sparse solution from a small number of measurements.

In the CS framework, $l_{p}$-norm (0<p<1) is the most effective measurement of the sparsity of a vector [16], so the corresponding $l_{p}$-norm minimization can more rapidly reconstruct a sparse solution with the less number of measurements than that of $l_{0}$-norm and $l_{1}$-norm minimization. Most current research work for CS-based target localization focuses on $l_{0}$-norm and $l_{1}$-norm minimization, however, very little work involves target localization via $l_{p}$-norm minimization. In this paper, different from existing work, we study $l_{p}$-norm minimization to solve CS-based target localization in WSNs. Because $l_{p}$-norm minimization is a non-convex optimization problem, the traditional recovery algorithms (e.g., the focal underdetermined system solver (FOCUSS) [17], the affine scaling algorithm [18] and the iteratively thresholding method (ITM) [19]) easily converge to suboptimal solutions. In fact, these solutions results in low-precision localization so that global optimization for $l_{p}$-norm minimization is very necessary to improve the position accuracy. For different application models, researchers have proposed many strategies (e.g., simulated annealing algorithm, neural networks, genetic algorithm and swarm algorithm) to solve global optimization, but these methods cannot be directly applied to $l_{p}$-norm minimization.

To this end, we propose a novel energy-level jumping (ELJ) algorithm to solve global optimization for $l_{p}$-norm minimization in CS-based target localization, which aims to obtain high-precision localization. Motivated by the concept of energy level in quantum mechanics, the ELJ algorithm assumes that the attraction basin of a local minimizer corresponds to an energy level. The iterative solution can jump out of the attraction basin of the current local minimizer by absorbing energy and enter another one with the lower energy level. By multiple energy-level jumps, the iterative solution enters the attraction basin with the lowest energy level such that we can easily obtain the globally optimal sparse solution via an existing recovery algorithm, and then precisely estimate the locations of targets.

Our work mainly focuses on the recovery algorithm for $l_{p}$-norm minimization in CS-based target localization and our major contributions are summarized as follows:
To reduce the number of measurements, we transform the CS-based target localization problem into $l_{p}$-norm minimization. Compared with the traditional CS-based target localization via $l_{0}$-norm or $l_{1}$-norm minimization, we provide a sparser solution via $l_{p}$-norm minimization, and then achieve more precise target localization.Inspired by the concept of energy level, we develop a novel ELJ algorithm to effectively solve the globally optimal sparse solution of $l_{p}$-norm minimization w hich corresponds to the most accurate locations of unknown targets. To the best of our knowledge, this is the first time to solve $l_{p}$-norm optimization by the idea of energy-level jumping. Furthermore, the theoretical analyses of the global convergence of our ELJ algorithm are provided to arouse a new and effective method, which can solve some practical non-convex optimization problems by piecewise way.The simulation results show that our ELJ algorithm can help some target localization algorithms to improve the position accuracy when these algorithms have to locate targets using suboptimal sparse solutions.

The rest of the paper is organized as follows. Section 2 briefly reviews the related works about target localization. Section 3 reviews the basic concepts of the CS theory and formulates the system model of CS-based target localization. In Section 4, the ELJ algorithm is derived to solve the globally optimal sparse solution of $l_{p}$-norm minimization. In Section 5, the global convergence analysis guarantees that our ELJ algorithm can accurately estimate the locations of targets using global minimizer. The simulation experiments in Section 6 validate the performance of our ELJ algorithm. Finally, Section 7 concludes this paper.

## 2. Related Work

In this section, we review the related works on some typical target localization algorithms in WSNs, then introduce some related works on CS-based target localization.

### 2.1. Typical Target Localization Algorithm in WSNs

By the type of the achieved measurements, the early target localization is classified into two main approaches: (a) triangulation and (b) fingerprinting. Triangulation approach utilizes the geometric property of triangle to locate targets, which mainly includes two methods: direction of arrival (DOA) and time of arrival (TOA). DOA estimates the angle from which a target emits the signal, but it cannot determine the accurate physical location of this target [20]. TOA estimates the arrival time of the signal from a target, but the measurement of the arrival time is highly affected by multipath and refraction [21]. The received signal strength (RSS)-based target localization system [22] seems to be more suit for WSNs because the values of RSS are available at the physical layer of each sensor without any additional hardware. Some hybrid schemes based on angle of arrival-time of arrival (AoA-ToA), time of arrival-received signal strength (ToA-RSS), and angle of arrival-received signal strength (AoA-RSS) signals are proposed to further enhance the localization performance [23,24]. To improve the location accuracy of hybrid (AoA-ToA) localization systems, a linear least squares (LLS) algorithm was used to obtain the location coordinates. Furthermore, a weighted linear least squares (WLLS) algorithm exhibited better localization performance than the LLS algorithm by utilizing the information present in the covariance of the incoming signals [25,26]. The authors of [27] proved that hybridization of different types of measurements (TOA/RSS) could enhance localization accuracy from Cramer-Rao lower bound (CRLB) analysis, but signal-to-noise ratio, path-loss exponent, as well as anchors placement all affected the hybrid TOA/RSS LLS localization accuracy in different ways. The authors of [28] applied a RSS/AOA hybrid positioning algorithm to increase the accuracy of localization when searching and rescuing the survivals in huge disaster.

There are many traditional methods to address RSS-based target localization which can be classified into two categories: the distance prediction-based and fingerprinting-based methods [29,30]. Under the complicated monitoring environment, a theoretical radio propagation model to characterize the distance between target and sensor node leads to high localization error. The fingerprinting-based method, however, can achieve acceptable localization results. More precisely, the scheme composes two phase: an offline phase, which collects the RSS measurements at some known locations in the monitoring area, and then stores them in a signature map; an online phase, which compares the current RSS measurements with the stored measurements in the signature map to obtain the locations of unknown targets [31]. To maintain the signature map, the fingerprinting-based method has to exchange a large number of data among sensors. Recently, many new target localization methods have been investigated without using signature map. The authors of [32] proposed the semantic localization algorithm to estimate the locations of targets, but knowing some previous conditions (e.g., the properties of the objects located in the proximity of a target) also results in a large amount of information exchange. The authors of [33] designed a multi-resolution strategy to address the estimation of the presence, position and posture of a target. To learn the unknown relation between the channel state information (CSI) string and the target state, the localization algorithm trained a set of support vector machine binary classifiers that spent extra computational overhead. Reducing information exchange and computational overhead is of great interest to develop alternative target localization methods.

### 2.2. CS-Based Target Localization Algorithm in WSNs

CS-based target localization has been proposed to tackle the above problem by recasting target localization into a sparse recovery problem. The authors of [34] pointed out that the locations of multiple targets being simultaneously determined was one of the significant advantages of CS-based localization methods while some traditional localization methods only took account of a single target. The authors of [35] provided a rigorous proof of the necessity of restricted isometry property (RIP) [36] for sparse recovery in the localization problem. Nevertheless, the measurements of sensors were coherent in spatial domain (i.e., the row vectors of a measurement matrix are coherent). A pre-processing procedure of orthogonalization [37,38] weakened the coherent of these row vectors in order to make the measurement matrix better satisfy RIP. The dynamic and vulnerable nature of radio signals caused the variation of measurements, so a dictionary refinement method was proposed to match the measurement matrix with the environmental change [39].Then, a variational expectation-maximization algorithm was adopted to realize joint dictionary refinement and sparse recovery.

As aforementioned, reconstructing a sparse solution from a small number of linear measurements is an essential issue to realize CS-based target localization. An improved GMP algorithm continuously divided the selected grid until to find the most likely locations of targets [40]. The authors of [41] noted that the transmitting powers of targets were different and unknown in real application, so they formulated the locations and transmitting powers of targets as a sparse vector. A basis pursuit algorithm SPGL1 [42] was used to simultaneously estimate the locations and transmitting powers of targets. However, the authors of [43] stated that the above localization algorithms needed to previously know the number of targets, which was usually unknown in practice. In addition, the relevant recovery algorithms easily converge to suboptimal solutions. In this paper, without any prior knowledge of the number of targets, we develop the ELJ algorithm to perform high-precision target localization by reconstructing the globally optimal sparse solution.

## 3. System Model of CS-Based Target Localization

In this section, we first review the CS theory to prepare for the system model of CS-based target localization, and then introduce the motivation for the ELJ algorithm.

### 3.1. Compressive Sensing

As we know many physical quantities are intrinsically sparse, so they can be represented by few nonzero expansion coefficients with respect to a suitable expansion basis [44,45,46,47]. On the basis of the sparsity, CS uses sub-sampling instead of Nyquist sampling to deal with a sparse or compressible signal such that it has attracted much attention in different fields (e.g., image processing [48], machine learning [49] and computer vision [50]). More precisely, assume that we can obtain the sparse representation of a signal $x\in R^{n}$ in the transform domain $\Psi \in R^{n\times n}$, i.e., x=Ψθ, where $\theta \in R^{n}$ is the coefficient vector with only *K* nonzero elements (i.e., θ is a *K*-sparse vector). In an encoder, instead of acquiring *n* samples, we sample *x* by the measurement matrix Φ and obtain a measurement vector $y\in R^{m}\left(m<n\right)$ as follows:
(1)y=ΦΨθ=Aθ,where $A\in R^{m\times n}$ is the CS matrix. In a decoder, we aims to reconstruct a sparse solution $\theta^{*}\in R^{n}$ and to obtain the reconstructed signal $x^{*}=\Psi \theta^{*}$. Due to the different norms, the authors of [51] indicated that the sparse recovery methods have three typical categories. First, a sparse solution can be reconstructed by the following $l_{0}$-norm minimization
(2)$$\min_{\theta }\left||\theta \right|{|}_{0}\;s.t.\;y=A\theta ,$$where *A* satisfies RIP. Some greedy algorithms (e.g., OMP, backtracking-based matching pursuit (BaMP) [52] and stagewise orthogonal matching pursuit (StOMP) [53]) are applied to solve Problem (2). These greedy algorithms, however, only choose *K* appropriate basis vectors to approximate *y* that may lead to locally optimal sparse solutions. Second, the following $l_{1}$-norm minimization is equivalent to $l_{0}$-norm minimization with high probability [13]:(3)$$\min_{\theta }\left||\theta \right|{|}_{1}\;s.t.\;y=A\theta .$$

Some effective algorithms such as BP, proximal algorithm [54] and dual augmented lagrangian method (DALM) [55] can reconstruct sparse solutions, but the solution of Problem (3) is often not as sparse as that of Problem (2). Third, some researchers investigate the following $l_{p}$-norm minimization
(4)$$\min_{\theta }\left||\theta \right|{|}_{p}^{p}\;s.t.\;y=A\theta .$$

Including iteratively reweighted $l_{1}$-norm minimization (IRL1) [56], iteratively reweighted least squares (IRLS) [36] and ITM, many recovery algorithms are proposed to solve non-convex optimization Problem (4), which has many local attraction basins. Without a good initial point or the sufficient number of measurements, these algorithms may achieve local minimizers rather than global minimizers. When obtaining a local minimizer, how to develop a feasible strategy to find the global minimizer is a very important problem for all applications of the CS technique. In this paper, we investigate global optimization in the background of CS-based target localization.

### 3.2. System Model and Algorithm Motivation

Assume that a WSN is composed of a fusion center and *m* randomly distributed sensor nodes. To apply the CS technique, the monitored area of WSNs is discretized into *n* grids. The locations of *K* targets in *n* grids are then represented by an *n*-dimensional vector *x* of which an entry is nonzero when its corresponding grid is occupied by a target. Since few targets are localized in the monitored area, the vector *x* is *K*-sparse. In this grid-based target localization model, we suppose that the target is located in the center of a grid [41]. According to the path loss model [57], the sensor node *i* senses the RSS from the target in the grid *j*, which follows
(5)$$p_{ij}=p_{0}-10\eta \log_{10}\left(d_{ij}/d_{0}\right),\;1\leq i\leq m,1\leq j\leq n,$$where $p_{ij}$ is the received signal power, $d_{ij}$ is the Euclidean distance, $d_{0}$ is a reference distance with the received power $p_{0}$, and η is the path loss coefficient with typical value between 2 and 4. If the RSS is affected by multi-path fading and shadowing effect in practice, some signal pre-processing methods including averaging multiple measurements and Kalman filtering can be applied to reduce the effect of measurement noise. The total RSS measured by *m* sensor nodes is then given by
(6)y=Px,where $y\in R^{m}$ is a measurement vector, $P\in R^{m\times n}$ is a measurement matrix, and $x\in R^{n}$ is an unknown *K*-sparse vector.

**Definition** **1.**
*Given m sensor nodes locate in a monitoring area divided into n grids, the CS-based target localization problem is transformed into the K-sparse signal recovery problem which is defined as follows:*
(7)$$\min_{x}E^{p}\left(x\right)=\left||x\right|{|}_{p}^{p}\;s.t.\;y=Px.$$


When solving Problem (7), the frequently-used algorithm IRLS is easy to obtain a locally optimal sparse solution. Therefore, making the iterative solution jump from the attraction basin of the current local minimizer to another one is a main challenge for global optimization of Problem (7). Our following work mainly focuses on solving this problem to precisely locate targets.

**Motivation for the Energy-Level Jumping algorithm.** Our work is motivated by the concept of energy level in quantum mechanics [58]. Figure 1 displays that an electron in atom can jump from an energy level to another one by emitting or absorbing a photon with the energy hv=hc/λ, where h is Planck’s constant, *c* is the speed of light and λ is wavelength. Meanwhile, every energy level has a ground state and some excited states, where some energy levels connect in certain excited states. If the absorbed energy is less than hv, the electron will transfer from the ground state to a excited state. According to this transition pattern of an electron among different energy levels, our sparse recovery algorithm for global optimization is discussed as follows.

## 4. Energy-Level Jumping Algorithm for CS-Based Target Localization

In this section, we elaborate our ELJ algorithm and present the advantage of target localization using the globally optimal sparse solution.

### 4.1. Preliminary Preparation

In order to better illustrate the relationship between the concept of energy level and our sparse recovery algorithm, we use an example to display the process of energy-level jumping for solving global optimization. Figure 2 shows that three attraction basins of the objective function correspond to three energy levels, where $x_{1}^{*}$ and $x_{g}^{*}$ are in the highest and lowest energy levels, respectively. First, IRLS is applied to obtain a locally optimal sparse solution $x_{1}^{*}$, and we try to make it move from the current attraction basin to another one with a lower energy level. Let $x_{1}^{*}$ absorb the energy $\varepsilon_{1}=r\cdot E^{\left(p\right)}\left(x_{1}^{*}\right)$ where *r* is a rate to measure the value of $\varepsilon_{1}$, then it jumps to a non-sparse solution $u_{1}^{0}$ in an excited state. Second, a non-sparse solution in the same excited state as $u_{1}^{0}$ needs to be found in another attraction basin. For this purpose, Figure 3 explains the connectivity of contour line, where the disconnected and connected contour lines are displayed, respectively. It is clear to see that a non-sparse solution $u_{1}^{*}$ can be searched along the connected contour line, and thus the energy-level jumping can be performed from $x_{1}^{*}$ to $u_{1}^{*}$. Nevertheless, we need to solve two problems in this process.
How do we obtain a non-sparse solution $u_{1}^{0}$?How do we construct a connected curve between two non-sparse solutions (i.e., $u_{1}^{0}$ and $u_{1}^{*}$) located in two attraction basins, respectively?

Without loss of generality, we suppose that a locally optimal solution $x_{l}^{*}$ with the energy $e_{l}$ is in the *l*-th energy level, where l∈{1,…,L}. The first problem is to find a non-sparse solution $u_{l}^{0}$ after $x_{l}^{*}$ absorbs the energy $\varepsilon_{l}=rE^{\left(p\right)}\left(x_{l}^{*}\right)$. For simplicity, we assume that the front *s* components of $x_{l}^{*}$ are zero, i.e.,
(8)$$x_{l}^{*}={\left(x_{l,1}^{*},x_{l,2}^{*}\right)}^{T}={\left(0,x_{l,2}^{*}\right)}^{T},$$where $x_{l,1}^{*}=\left(x_{l}^{*}\left(1\right),\cdots ,x_{l}^{*}\left(s\right)\right),x_{l,2}^{*}=\left(x_{l}^{*}\left(s+1\right),\cdots ,x_{l}^{*}\left(n\right)\right)$. By partitioning $P=(P_{1},P_{2})$ and setting $u_{l,2}^{0}=x_{l,2}^{*}$, we need to adjust the zero components of $x_{l}^{*}$ to $u_{l,1}^{0}$ satisfying
(9)$$\left\{\begin{array}{c} P_{1}u_{l,1}^{0}=0\\ E^{\left(p\right)}\left(u_{l,1}^{0}\right)=\varepsilon_{l} \end{array}\right.,$$where $u_{l}^{0}={\left(u_{l,1}^{0},u_{l,2}^{0}\right)}^{T}={\left(u_{l}^{0}\left(1\right),\cdots ,u_{l}^{0}\left(s\right),u_{l}^{0}\left(s+1\right),\cdots ,u_{l}^{0}\left(n\right)\right)}^{T}$. To this purpose, we randomly choose a nonzero solution *v* of the linear equations $P_{1}v=0$. If $u_{l,1}^{0}={\left(\frac{\varepsilon_{l}}{E^{\left(p\right)}\left(v\right)}\right)}^{\frac{1}{p}}v$, then $x_{l}^{*}$ is transformed into $u_{l}^{0}={\left(u_{l,1}^{0},x_{l,2}^{*}\right)}^{T}$ with the energy $e_{l}+\varepsilon_{l}$.

After $x_{l}^{*}$ successfully jumps to $u_{l}^{0}$, the second problem is to find $u_{l}^{*}$ along the connected contour line. In practice, it is difficult to describe the contour line of an objective function, especially for a large-scale optimization problem. In next section, we will show how the homotopy method [59] constructs a connected curve between $u_{l}^{0}$ and $u_{l}^{*}$.

### 4.2. Homotopy Curve Construction

To construct a homotopy curve between $u_{l}^{0}$ and $u_{l}^{*}$, we first guarantee the existence of $u_{l}^{*}$ in term of the following lemma.

**Lemma** **1.**
*Assume that two locally optimal sparse solutions $x_{l}^{*}$ and $x_{l+1}^{*}$ have the energy $e_{l}$ and $e_{l+1}\left(e_{l+1}<e_{l}\right)$, respectively. Given the energy $\varepsilon_{l}$, there exists a non-sparse solution $u_{l}^{*}$ with the energy $e_{l}+\varepsilon_{l}$ in the attraction basin of $x_{l+1}^{*}$.*


**Proof.** For convenience, we suppose that the front part of $x_{l+1}^{*}$ is zero, i.e.,
(10)$$x_{l+1}^{*}={\left(x_{l+1,1}^{*},x_{l+1,2}^{*}\right)}^{T}={\left(0,x_{l+1,2}^{*}\right)}^{T},$$By partitioning $P=(P_{1},P_{2})$ and setting $u_{l,2}^{*}=x_{l+1,2}^{*}$, we try to find a non-sparse solution $u_{l}^{*}={\left(u_{l,1}^{*},u_{l,2}^{*}\right)}^{T}$ in the attraction basin of $x_{l+1}^{*}$, where $u_{l,1}^{*}$ needs to satisfy
(11)$$\left\{\begin{array}{c} P_{1}u_{l,1}^{*}=0\\ E^{\left(p\right)}\left(u_{l,1}^{*}\right)=e_{l}+\varepsilon_{l}-e_{l+1} \end{array}\right.,$$Randomly choosing a nonzero solution $\tilde{v}$ of the linear equations $P_{1}\tilde{v}=0$ and computing $u_{l,1}^{*}={\left(\frac{e_{l}+\varepsilon_{l}-e_{l+1}}{E^{\left(p\right)}\left(\tilde{v}\right)}\right)}^{\frac{1}{p}}\tilde{v}$, we can find a non-sparse solution $u_{l}^{*}={\left(u_{l,1}^{*},x_{l+1,2}^{*}\right)}^{T}$ with the energy $e_{l}+\varepsilon_{l}$ in the attraction basin of $x_{l+1}^{*}$. □

Based on Lemma 1, we solve $u_{l}^{*}$ by the following nonlinear equations
(12)$$\left\{\begin{array}{c} Pu-y=0\\ \sum_{i=1}^{n}|u\left(i\right){|}^{p}-\left(e_{l}+\varepsilon_{l}\right)=0 \end{array}\right.,$$Applying the homotopy method to solve Problem (12), we define a homotopy map *H* satisfying
(13)$$\left\{\begin{array}{c} H(u,t)=tF(u)+(1-t)G(u)=0\\ H(u,0)=G(u)\\ H(u,1)=F(u) \end{array}\right.,$$where $F\left(u\right)=\left(\begin{array}{c} Pu-y\\ \sum_{i=1}^{n}|u\left(i\right){|}^{p}-\left(e_{l}+\varepsilon_{l}\right) \end{array}\right)$, $\rho_{1}\left(u,u_{l}^{0}\right)={\left(u\left(1\right)-u_{l}^{0}\left(1\right),\cdots ,u\left(m\right)-u_{l}^{0}\left(m\right)\right)}^{T}$,

$G\left(u\right)=\left(\begin{array}{c} \rho_{1}\left(u,u_{l}^{0}\right)\\ \rho_{2}\left(u,u_{l}^{0}\right) \end{array}\right)$, $\rho_{2}\left(u,u_{l}^{0}\right)={\left(u\left(m+1\right)-u_{l}^{0}\left(m+1\right)\right)}^{2}+\cdots +{\left(u\left(n\right)-u_{l}^{0}\left(n\right)\right)}^{2}$.

Now, Problem (13) is transformed into the following initial value problem of differential equations to establish a homotopy curve u=u(t):
(14)$$\left\{\begin{array}{c} \frac{\partial H}{\partial u}\frac{du}{dt}+\frac{\partial H}{\partial t}=0\\ u\left(0\right)=u_{l}^{0} \end{array}\right.,$$where $\frac{\partial H}{\partial u}=\left(\begin{array}{cccccc} tp_{11}+1-t & \cdots & tp_{1m} & tp_{1,m+1} & \cdots & tp_{1n}\\ \vdots & & \vdots & & \vdots & \\ tp_{m1} & \cdots & tp_{mm}+1-t & tp_{m,m+1} & \cdots & tp_{mn}\\ \alpha_{1} & \cdots & \alpha_{m} & \beta_{m+1} & \cdots & \beta_{n} \end{array}\right)$,

$\frac{\partial H}{\partial t}=\left(\begin{array}{c} Pu-y-\rho_{1}\left(u,u_{l}^{0}\right)\\ \sum_{i=1}^{n}|u\left(i\right){|}^{p}-(e_{l}+\varepsilon_{l})-\rho_{2}\left(u,u_{l}^{0}\right) \end{array}\right)$, $\alpha_{i}=p|u\left(i\right){|}^{p-2}u\left(i\right)\left(i=1,\cdots ,m\right)$,

$\beta_{i}=\alpha_{i}+\left(2-2t\right)\left(u\left(i\right)-u_{l}^{0}\left(i\right)\right)\left(i=m+1,\cdots ,n\right)$.

Going back to see the example in Figure 3, we note that Problem (14) has infinitely solutions in the connected contour line. Our work is to find one of solutions $u_{1}^{*}$ in an attraction basin with the lower energy level. Hence we need to distinguish $u_{1}^{0}$ and $u_{1}^{*}$, otherwise $u_{1}^{0}$ cannot jump from an energy level to another one. It is observed that $u_{1}^{0}$ being close to $x_{1}^{*}$ is nearly sparse (i.e., some of components approach zero) while $u_{1}^{*}$ being away from $x_{2}^{*}$ is non-sparse. Thus, $l_{2}$-norm is used to measure the non-sparsity of two vectors to distinguish them. For example, a nearly sparse solution $u_{1}^{0}={\left(0.9500,0.0001,0.8779\right)}^{T}$ and a non-sparse solution $u_{1}^{*}={\left(0.8,0.1,0.3\right)}^{T}$ locate in two different energy levels, we have $\left|\right|u_{1}^{*}{\left|\right|}_{0.4}^{0.4}=\left|\right|u_{1}^{0}{\left|\right|}_{0.4}^{0.4}$ and $\left|\right|u_{1}^{*}{\left|\right|}_{2}^{2}<||u_{1}^{0}{\left|\right|}_{2}^{2}$.

As discussed, if we apply the homotopy method to generate a sequence ${\left\{u_{l}^{j}\right\}}_{j=1}^{J}$, where each solution $u_{l}^{j}$ satisfies
(15)$$\min||u_{l}^{j}{\left|\right|}_{2}^{2},$$then the limit of this sequence $u_{l}^{*}$ will be in an attraction basin with the lower energy level. Furthermore, Problem (15) is used to compute the optimal stepsize of each iteration by the following modified Euler’s forecast-Newton correction homotopy method [60]. The iterative process is summarized as follows:
(16)$$\begin{array}{c} {\tilde{u}}_{l}^{j+1}=u_{l}^{j}-{\left(\frac{\partial H(u_{l}^{j},t_{l}^{j})}{\partial u}\right)}^{+}\frac{\partial H(u_{l}^{j},t_{l}^{j})}{\partial t}\Delta t_{l}^{j+1},\;\mathrm{where}\;\Delta t_{l}^{j+1}=\min||{\tilde{u}}_{l}^{j+1}{\left|\right|}_{2}^{2}\\ u_{l}^{j+1}={\tilde{u}}_{l}^{j+1}-{\left(\frac{\partial H({\tilde{u}}_{l}^{j+1},t_{l}^{j})}{\partial u}\right)}^{+}H\left({\tilde{u}}_{l}^{j+1},t_{l}^{j}\right) \end{array}$$where $t_{l}^{0}=0<\cdots <t_{l}^{j}<\cdots <t_{l}^{J}=1$ and the optimal stepsize $\Delta t_{l}^{j+1}=t_{l}^{j+1}-t_{l}^{j}$. We thus construct a homotopy curve between ${\left(u_{l}^{0},0\right)}^{T}$ and ${\left(u_{l}^{*},1\right)}^{T}$ using the sequence ${\left\{\left(u_{l}^{j},t_{l}^{j}\right)\right\}}_{j=0}^{J}$.

Once a homotopy curve is constructed in two attraction basins, the iterative solution has an opportunity to jump from a high energy level to a low one, and then the global convergence of $l_{p}$-norm minimization can be realized step by step. Figure 4 gives an example to display a process of global convergence in the contour map. In the first step, a local optimal sparse solution $x_{1}^{*}$ is solved by IRLS, and it jumps to $u_{1}^{0}$ after absorbing the energy $\varepsilon_{1}$. A homotopy curve composed of the sequence ${\left\{\left(u_{1}^{j},t_{1}^{j}\right)\right\}}_{j=0}^{2}$ is estabished between $u_{1}^{0}$ and $u_{1}^{*}$. Along this homotopy curve, $u_{1}^{*}$ enters another attraction basin with the lower energy level. In the second step, using $u_{1}^{*}$ as an initial point, IRLS is reapplied to obtain a sparser solution $x_{2}^{*}$ of Problem (7). If $x_{2}^{*}$ is still a local minimizer, it jumps out of the current attraction basin along the homotopy curve between ${\left(u_{2}^{0},t_{2}^{0}\right)}^{T}$ and ${\left(u_{2}^{*},t_{2}^{3}\right)}^{T}$. Starting from $u_{2}^{*}$, we reapply IRLS to find the globally optimal sparse solution $x_{g}^{*}$. According to the above explanation, the global convergence is accomplished by two energy-level jumps $x_{1}^{*}\to x_{2}^{*}\to x_{g}^{*}$.

It is observed that the number of sparse solutions of Problem (7) is finite and the number of the corresponding energy levels is finite. Meanwhile, if starting from the least squares solution, the locally optimal sparse solution solved by IRLS is usually close to the globally optimal sparse solution. Therefore, the global minimizer of Problem (7) can be found within a small number of energy-level jumps. On the other hand, any of the traditional or state-of-the-art recovery algorithms can replace IRLS to solve a local minimizer that has little impact on the algorithm structure of ELJ. By constructing a connected homotopy curve between the attraction basins of two locally optimal sparse solutions, our ELJ algorithm provides a new idea to effectively solve the non-convex optimization problem.

### 4.3. CS-Based Target Localization via Energy-Level Jumping Algorithm

Once obtaining the globally optimal sparse solution of Problem (7), the grid locations of targets are determined by
(17)$$id=\{i|x_{g}^{*}\left(i\right)\neq 0,i=1,\cdots ,n\},$$and then the center of a grid is the estimated location of a target. Next, CS-based target localization via our ELJ algorithm is summarized as follow.

Algorithm 1 displays that the iterative process is stopped by comparing the energy values of two locally optimal sparse solutions, so the sparse reconstruction of Problem (7) does not need any prior knowledge of the sparsity of the solution. This advantage is beneficial to apply our ELJ algorithm to some practical localization scenarios in centralized setup, which have no prior knowledge of the number of targets. On the other hand, our ELJ algorithm can also be applied to a distributed scenario. For example, a clusterhead can locate targets in its cluster by applying the ELJ algorithm after it receives the sufficient number of measurements from its member nodes.
**Algorithm 1:** Compressive sensing-based target localization via energy-level jumping algorithm **Input:** A measurement matrix *P*, a measurement vector *y*, an error threshold δ. **Initialize:** An initial point $x_{0}=P^{+}y$, an iterative index *l*. **while** Stopping criterion not met do 1: Apply IRLS to reconstruct a locally optimal sparse solution $x_{l}^{*}$. 2: Let $x_{l}^{*}$ jump to $u_{l}^{0}$ by absorbing the energy $\varepsilon_{l}$. 3: Apply the modified Euler’s forecast-Newton correction homotopy method to construct a homotopy curve between $u_{l}^{0}$ and $u_{l}^{*}$. 4: Update $x_{0}=u_{l}^{*}$ and apply IRLS to find a sparser solution $x_{l+1}^{*}$. 5: Increase iterative index *l*. **end while** if $|E^{\left(p\right)}\left(x_{l+1}^{*}\right)-E^{\left(p\right)}\left(x_{l}^{*}\right)|<\delta$. **Output:** 1: The globally optimal sparse solution $x_{g}^{*}=x_{l+1}^{*}$. 2: The grid locations of targets
$id=\{i|x_{g}^{*}\left(i\right)\neq 0,i=1,\cdots ,n\}.$

## 5. Convergence Analysis

From the above discussion, we see that the position accuracy depends on the convergence property of the sparse recovery algorithm. In this section, we turn to discuss the global convergence behavior of our ELJ algorithm in order to verify the feasibility of high-precision target localization.

### 5.1. Convergence of Our Homotopy Method

To realize energy-level jumping, a homotopy curve must be established between two attraction basins. The convergence behavior of our homotopy method is discussed as follows.

**Definition** **2**([59])**.** *If $f:R^{n}\to R^{m}$ is a smooth map and the Jacobi matrix $\frac{\partial f}{\partial u}\left(u_{0}\right)$ is full row rank, then $u_{0}$ and $y_{0}=f\left(u_{0}\right)$ are called a regular point and a regular value, respectively.*

**Definition** **3**([59])**.** *Given $f:R^{n}\to R^{m}$ is a smooth map and zero is its regular value, we have a fix-point homotopy H(u,t)=tf(u)+(1−t)g(u), where $g\left(u\right)=u-u_{0}$.*

**Definition** **4**([59])**.** *If zero is a regular value of H(u,t) for any $u_{0}\in R^{n}$, there exists a homotopy curve between ${\left(u_{0},0\right)}^{T}$ and ${\left(u^{*},1\right)}^{T}$ in the zero set*
(18)$$H^{-1}\left(0\right)=\left\{{\left(u,t\right)}^{T}|H\left(u,t\right)=0\right\},$$*where $f(u^{*})=0$.*

**Theorem** **1.**
*F(u) in (13) is a smooth map and zero is its regular value. If zero is a regular value of H(u,t) in (13) for a given $u_{l}^{0}\in R^{n}$, there exists a homotopy curve between ${\left(u_{l}^{0},0\right)}^{T}$ and ${\left(u_{l}^{*},1\right)}^{T}$ in the zero set*
(19)$$H^{-1}\left(0\right)=\left\{{\left(u,t\right)}^{T}|H\left(t,u\right)=tF\left(u\right)+\left(1-t\right)G\left(u\right)=0\right\},$$
*where $F(u_{l}^{*})=0$.*


**Proof.** From (13), zero is a regular value of F(u) because of $F(u_{l}^{0})=0$. Furthermore, the Jacobi matrix of H(u,t) with respect to $\left(u,u_{l}^{0},t\right)$ is calculated as follows:
(20)$$\frac{\partial H}{\partial (u,u_{l}^{0},t)}=\left(\frac{\partial H}{\partial u},\frac{\partial H}{\partial u_{l}^{0}},\frac{\partial H}{\partial t}\right),$$where $\frac{\partial H}{\partial u_{l}^{0}}=\left(\begin{array}{cccc} t-1 & \cdots & 0 & \begin{array}{ccc} 0 & \cdots & 0 \end{array}\\ \vdots & & \vdots & \begin{array}{ccc} \vdots & & \vdots \end{array}\\ 0 & \cdots & t-1 & \begin{array}{ccc} 0 & \cdots & 0 \end{array}\\ 0 & \cdots & 0 & \begin{array}{ccc} \gamma_{m+1} & \cdots & \gamma_{n} \end{array} \end{array}\right)$ and $\gamma_{i}=-2\left(u\left(i\right)-u_{l}^{0}\left(i\right)\right)\left(i=m+1,\cdots ,n\right).$ □

The row rank of $\frac{\partial H}{\partial (u,u_{l}^{0},t)}$ is m+1, so the Jacobi matrix of H(u,t) is full row rank, i.e., zero is a regular value of H(u,t). According to Lemma 2, the zero set $H^{-1}\left(0\right)$ has a homotopy curve between ${\left(u_{l}^{0},0\right)}^{T}$ and ${\left(u_{l}^{*},1\right)}^{T}$.

The convergence property of our homotopy method in Theorem 1 is one of the important parts of the global convergence of our ELJ algorithm.

### 5.2. Global Convergence of Energy-Level Jumping Algorithm

We use IRLS to find a local minimizer, so the convergence of IRLS is another important part of the global convergence of our ELJ algorithm. The convergence theorem of IRLS [18] is reviewed as follows.

**Theorem** **2.**
*Let $x^{*}$ be a sparse solution of Problem (7). Starting from an initial point $x_{0}$, IRLS generates a sequence ${\left\{x_{k}\right\}}_{k=1}^{\infty }$ converging to $x^{*}$. The global rate of convergence is order 2−p.*


Based on Theorem 1 and Theorem 2, we provide the following global convergence of our ELJ algorithm.

**Theorem** **3.**
*Let $x_{g}^{*}$ be the globally optimal sparse solution of Problem (7). For any initial point $x_{0}$, ELJ generates a sequence ${\left\{x_{l}^{*}\right\}}_{l=1}^{L}$ converging to $x_{g}^{*}$, where $x_{l}^{*}$ is a locally optimal sparse solution solved by IRLS.*


**Proof.** To show the global convergence of the ELJ algorithm, we consider the conditions of the global convergence theorem [18]. □

(1)*Solution set*. A set Λ is obtained by collecting all sparse solutions of Problem (7), i.e.,
(21)$$\Lambda =\{x:K\leq ||x|{|}_{0}<n,y=Px\}.$$The set (21) has a bounded subset $\Lambda_{b}={\left\{x_{l}^{*}\right\}}_{l=1}^{L}$, where $x_{l}^{*}$ is a locally optimal sparse solution solved by IRLS. The limit of ${\left\{x_{l}^{*}\right\}}_{l=1}^{L}$ exists due to the bounded convergence theorem.(2)*Descent function*. Note that the objective function $E^{\left(p\right)}\left(x\right)$ is a descent function. Corresponding to the sequence ${\left\{x_{l}^{*}\right\}}_{l=1}^{L}$, there exists a monotonically decreasing energy sequence ${\left\{E^{\left(p\right)}\left(x_{l}^{*}\right)\right\}}_{l=1}^{L}$ such that
(22)$$E^{\left(p\right)}\left(x_{g}^{*}\right)=E^{\left(p\right)}\left(x_{L}^{*}\right)<E^{\left(p\right)}\left(x_{L-1}^{*}\right)<\cdots <E^{\left(p\right)}\left(x_{1}^{*}\right)$$Hence, ELJ is a descent algorithm and the expression of global convergence is $\lim_{l\to L}x_{l}^{*}=x_{g}^{*}$.(3)*The rate of piecewise convergence*. When we investigate the global minimizer, the solving process is divided into two stages: our modified Euler’s forecast-Newton correction homotopy method is used to find an initial point of local optimization, and IRLS is used to find a local minimizer. Our homotopy method is linear convergence while Theorem 2 states that the global convergence rate of IRLS is order 2−p. Hence, the convergence rate of ELJ is piecewise.

Theorem 3 reveals that the global convergence of our ELJ algorithm is related to the local optimization algorithm. If applying different recovery algorithms (e.g., IRL1, IRLS and ITM) to find a local minimizer of Problem (7), the number of energy-level jumps can be used to evaluate the convergence efficiency of ELJ in terms of the following analysis. According to the convergence Theorem 3, the solving process of ELJ is divided into two stages. In first stage, a recovery algorithm is applied to find a locally optimal sparse solution, which is limited by the initial point $x_{0}$. In second stage, no matter what kinds of recovery algorithms we use, the same homotopy method is used to make the locally sparse optimal solution jump from the current attraction basin to another one. This means that our homotopy method has little impact on the number of energy-level jumps. Therefore, the number of energy-level jumps of our ELJ algorithm mainly depends on the initial point $x_{0}$ and the used recovery algorithm.

## 6. Simulation Results and Analysis

The performance of our ELJ algorithm for global optimization in CS-based target localization is studied and analyzed by the following simulations.

### 6.1. Numerical Example

We firstly give the following numerical example to validate the feasibility of our ELJ algorithm.
(23)$$\min_{x}E^{\left(0.5\right)}\left(x\right)=\sum_{i=1}^{8}\;|x\left(i\right){|}^{0.5}\;s.t.\;y=Ax,$$where $A=\left(\begin{array}{cccccccc} 1 & 1 & -3 & 6 & 1 & 5 & 4 & 3\\ 0 & 1 & 2 & 1 & 2 & 3 & 1 & -2\\ 11 & 2 & -5 & 1 & -2 & 1 & 0 & -2 \end{array}\right)$ and $y={\left(3.7,0.7,2.5\right)}^{T}$.

To solve problem (23), IRLS is used to find a locally optimal sparse solution $x_{1}^{*}=$(0.1730,0,0,0.5447,0,0.0518,0,0)${}^{T}$ with the energy $e_{1}=$1.5860. By absorbing the energy $\varepsilon_{1}=0.63e_{1}$, $x_{1}^{*}$ jumps to $u_{1}^{0}={\left(0.1730,0.0186,0.0287,0.5447,-0.0530,0.0518,0.0301,0.0009\right)}^{T}$. Our homotopy method establishes a curve between $u_{1}^{0}$ and $u_{1}^{*}={\left(-0.0011,0.0081,-0.3215,0.1465,-0.0951,0.4015,-0.0035,0.0023\right)}^{T}$ with the energy $E^{\left(p\right)}\left(u_{1}^{*}\right)=$2.5860, so $u_{1}^{*}$ jumps out of the attraction basin of $x_{1}^{*}$. From $u_{1}^{*}$, IRLS is reused to find the globally optimal sparse solution $x_{g}^{*}={\left(0,0,-0.4,0,0,0.5,0,0\right)}^{T}$. Global optimization for Problem (23) is accomplished by two energy-level jumps. Furthermore, our ELJ algorithm can also obtain the global minimizer if the absorbing energy $\varepsilon_{1}$ is $0.3e_{1},0.4e_{1}$ or $0.5e_{1}$. The rule of choosing suitable absorbing energy is investigated as follows.

Given a measurement matrix $P\in R^{60\times 256}$ and a *K*-sparse vector $x_{g}^{*}$, the measurement vector is set by $y=Px_{g}^{*}$. The probability of finding the global minimizer is defined as $p_{r}=N_{g}/N_{t}$, where $N_{g}$ is the total number of experiments for obtaining the global minimizers and $N_{t}$ is the total number of experiments. Applying the frequently used IRL1, IRLS and ITM to search a local minimizer, Figure 5 shows that ELJ has the high probability to obtain the global minimizer when the rate *r* is chosen from [0.3, 0.55]. However, our ELJ algorithm cannot find the global minimizer if the absorbing energy is less than $0.3E^{\left(p\right)}\left(x_{l}^{*}\right)$. This is because $u_{l}^{0}$ is on the disconnected contour line, and our homotopy method cannot find $u_{l}^{*}$. We obtain the similar result when the absorbing energy is larger than $0.55E^{\left(p\right)}\left(x_{l}^{*}\right)$. The reason is that $u_{l}^{0}$ is on the connected contour line with high energy and $u_{l}^{*}$ enters an attraction basin with the higher energy level. Accordingly, the energy of the reconstructed solution $x_{l+1}^{*}$ is higher than that of $x_{l}^{*}$, i.e., the sparser locally optimal solution cannot be found. To make ELJ become a descent algorithm due to Theorem 3, we prefer to choose *r* from [0.3, 0.5]. In fact, any value from 0.3 to 0.5 can be assigned to *r* without affecting the process of energy-level jumping, because the connected contour lines of the objective function between two attraction basins are infinite in Figure 3. In the following experiments, we all set r=0.4.

### 6.2. Accuracy of Target Localization

In our simulations, an 256 m${}^{2}$ area is divided into 16×16 grids, where K=5 targets are randomly deployed without knowing their locations and m=30 sensor nodes collect the compressive measurements. We set the system parameters as follows: the path loss coefficient η=2 and the reference distance $d_{0}=1$ with the received power $p_{0}=-40$ dBm. Averaging the Euclidean distance between the estimated locations and the actual locations of *K* targets denotes localization error
(24)$$Err=\frac{1}{K}\sum_{i=1}^{K}\sqrt{{\left(\xi_{i}-{\hat{\xi }}_{i}\right)}^{2}+{\left(\zeta_{i}-{\hat{\zeta }}_{i}\right)}^{2}}$$where $\left(\xi_{i},\zeta_{i}\right)$ and $\left({\hat{\xi }}_{i},{\hat{\zeta }}_{i}\right)$ are the actual and estimated locations of the *i*-th target.

Figure 6 shows that the localization via ELJ is very precise while the localization via BP, OMP, GMP and IRL1 has high estimation error. This is because $l_{p}$-norm minimization can provide the sparser solution than that of $l_{0}$-norm and $l_{1}$-norm minimization, and ELJ uses the global minimizer to obtain the accurately estimated locations of targets while BP, OMP, GMP and IRL1 use the local minimizer to locate targets leading to the missed targets and the false targets. Furthermore, the localization time of BP, OMP, GMP, IRL1 and ELJ 0.4584 s, 0.0487 s, 0.7874 s, 0.0808 s and 1.1099 s, respectively. Although the localization time of four algorithms is less than that of ELJ, they provide incorrectly estimated locations of some of targets. It is observed that the performance of the ELJ algorithm is a trade-off between localization accuracy and localization time. If the primary aim of CS-based target localization is to determine the locations of targets with high precision, we need to tolerant the slight increasing of computation time of ELJ compared to that of BP, OMP, GMP and IRL1. On the other hand, satisfying the real-time requirement of CS-based target localization is at the cost of localization accuracy of ELJ, namely we only find a better locally optimal sparse solution rather than the globally optimal sparse solution to locate targets in limited localization time.

### 6.3. Influence of the Number of Measurements

As we know increasing the number of measurements can improve the reconstruction performance of BP, OMP, GMP and IRL1. Fixing the number of targets K=5, we vary the number of sensor nodes from 20 to 80 to compare the localization performance of five recovery algorithms. The mean localization error $\bar{Err}=\frac{1}{N_{t}}\sum_{i=1}^{N_{t}}Err_{i}$ is used to evaluate localization precision, where $Err_{i}$ is the localization error of the *i*-th experiment. In Figure 7a, the mean localization error of BP, OMP, GMP, IRL1 and ELJ decreases with the increasing of the number of sensor nodes (i.e., the number of measurements). To achieve accurate sparse solutions, the number of measurements of BP, OMP and GMP is more than 60 (i.e., m≥12K), which is similar to the experimental results in [37]. IRL1 cannot accurately determine the locations of targets when the number of measurements is less than 40. As can be seen, five targets are accurately located as the number of measurements is more than 25 (i.e., m≥5K), so the target localization via ELJ achieves the best performance as well as greatly reduces the sensing consumption of sensors. Figure 7b displays that the corresponding mean localization time of BP, OMP, GMP, IRL1 and ELJ increases with the increasing of the number of measurements. ELJ spends more mean computation time than other four recovery algorithms to accomplish the higher precision localization because it performs the multiple number of local optimization.

### 6.4. Time Complexity Analysis

The above experiments explain that the ELJ algorithm improves the localization accuracy at the cost of increasing the computation time. To reduce the computational overhead of localization, it is necessary to further analyze the reconstruction performance of ELJ. We define the mean number of energy-level jumps $\bar{N}=\frac{1}{N_{t}}\sum_{i=1}^{N_{t}}N_{i}$ to evaluate the time complexity of ELJ, where $N_{i}$ is the number of energy-level jumps in the *i*-th experiment and $N_{t}=$ 50 is the total number of experiments. Figure 8a,b show that the mean number of energy-level jumps of five recovery algorithms and the corresponding mean localization time all increase when the number of targets increases, especially locating the larger number of targets leads to high time complexity. At the moment we need more measurements to acquire the location information of these targets in order to reduce the time consumption of target localization. For example, the mean localization time with respect to m=8K reduce at most 251% compared to m=5K in Figure 8b. On the other hand, Figure 8c shows that the larger *p* is, the more rapidly ELJ finds the global minimizer. We prefer to choose *p* from [0.6, 1), thereby reducing the mean energy-level jumps. Accordingly, the mean localization time as p=0.8 reduces from 69.44% to 109.21% compared to p=0.2 in Figure 8d. The results in Figure 8a–d imply that increasing the number of measurements or decreasing the non-convex property of $l_{p}$-norm can effectively improve the localization performance of ELJ. In practice, we can pre-set a larger value of *p* to accelerate the global convergence of the ELJ algorithm.

### 6.5. Influence of Local Recovery Algorithm

IRL1 and ITM also are the local recovery algorithms of $l_{p}$-norm optimization, so we need to investigate the localization performance when ELJ uses IRL1, IRLS and ITM (i.e., ELJ-IRL1, ELJ-IRLS and ELJ-ITM) to solve local minimizer, respectively. In Figure 9a, the mean number of energy-level jumps of ELJ-ITM is more than that of ELJ-IRL1 and ELJ-IRLS when p=0.5 and p=0.9. Meanwhile, Figure 9b shows that ELJ-ITM needs more mean localization time than that of ELJ-IRL1 and ELJ-IRLS. The similar results are showed in Figure 9c,d when m=6K and m=7K. Here, the experimental results reveal that our ELJ algorithm can obtain the better localization performance when using IRL1 and IRLS to solve local optimization.

### 6.6. Influence of Measurement Noise

The complex and volatile environment leads to the measurement noise including the measured deviation in the path loss model and the measured deviation of compressive sampling, which is set as Gaussian white noise in our experiments. Recovering sparse solution from the noisy measurements may cause the localization error, so we test the robustness of ELJ under the different measurement noise levels quantified by Signal-to-Noise-Ratio (SNR). We deploy 10 targets and 65 sensors in the monitoring area to examine the localization accuracy for the different noise levels, where BP, OMP, GMP, IRL1 and ELJ are applied to solve Problem (7). When the noise levels vary from −5 to 25 dB, Figure 10a shows that our ELJ algorithm can tolerate some degrees of measurement noise while the high measurement noise has great impact on the mean localization error of BP, OMP, GMP and IRL1. It is clear that the robustness of ELJ is better than BP, OMP, GMP and IRL1. Under low SNR, the localization precision of ELJ outperforms that of BP, OMP, GMP and IRL1 from 248-fold to 380-fold. These results reveal that ELJ has the best performance of noise reduction at the cost of spending more localization time showed in Figure 10b.

## 7. Conclusions

In this paper, we propose a novel ELJ algorithm for global optimization to realize high-precision target localization. The proposed algorithm makes the iterative solution jump out of the attraction basin of a local minimizer and enter another attraction basin with the lower energy level. A sequence of locally optimal sparse solutions with the monotonous decreasing energy levels converges to the globally optimal sparse solution. The detailed convergence analysis guarantees the global convergence of the proposed algorithm. Our simulation results show that the ELJ algorithm can provide the more accurately estimated locations of targets compared to other existing sparse recovery algorithms. In the future work, we will focus on the following work:
Various types of propagation models replacing the path loss model will be applied to test the real localization performance of the proposed algorithm.Installing a WSN in a smart community, various types of application scenarios (e.g., CS-based target localization for people and car) will be considered to test the practicability of the proposed algorithm such that the parameters are more reasonably set.To reduce localization delay, the ELJ algorithm will be improved by accelerating global convergence and reducing computation time.

## Figures and Tables

**Figure 1 sensors-19-02502-f001:**
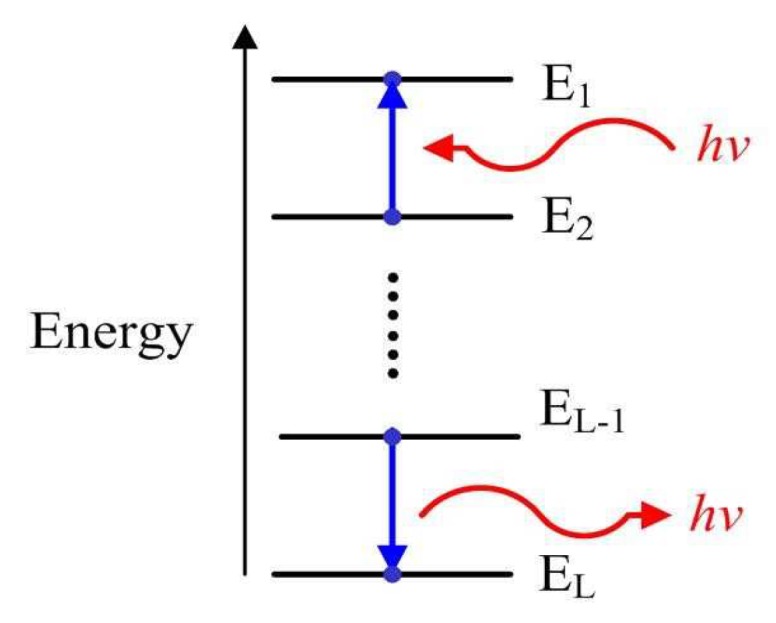
The concept of energy level in quantum mechanics. The electron in atom increases energy level from $E_{2}$ to $E_{1}$ by absorbing a photon or decreases energy level from $E_{L-1}$ to $E_{L}$ by emitting a photon, where the energy of the photon is hv.

**Figure 2 sensors-19-02502-f002:**
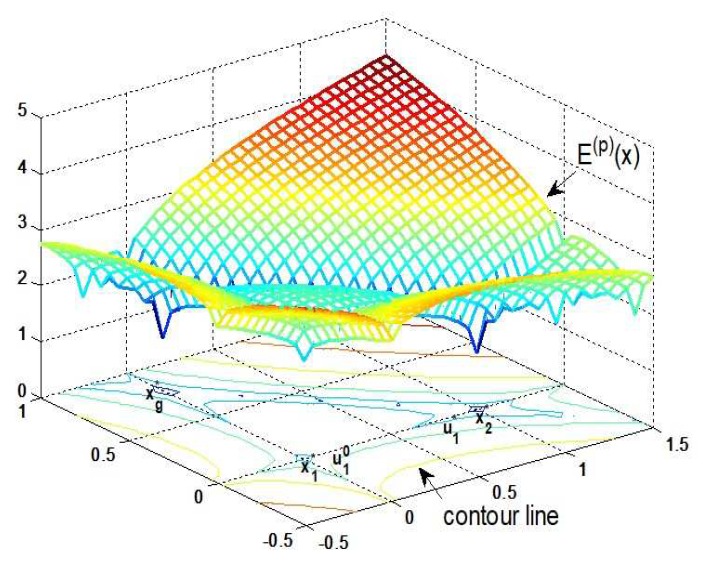
Three attraction basins of the objective function correspond to three energy levels in 3-D space.

**Figure 3 sensors-19-02502-f003:**
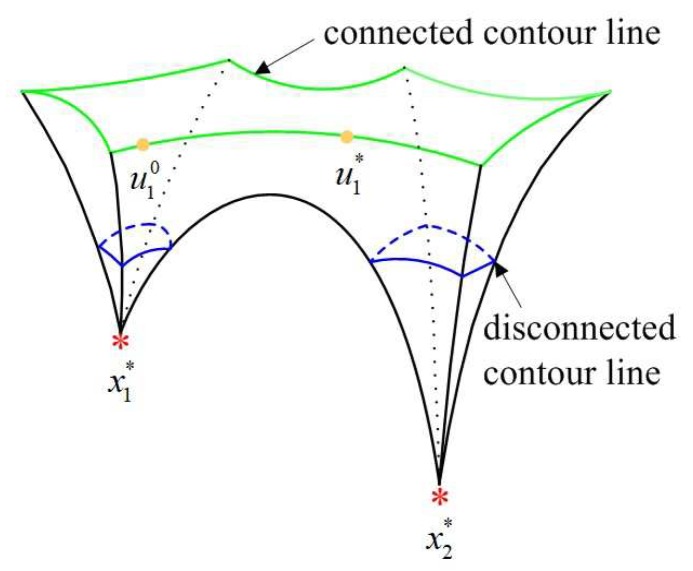
Connected and disconnected contour lines of the objective function.

**Figure 4 sensors-19-02502-f004:**
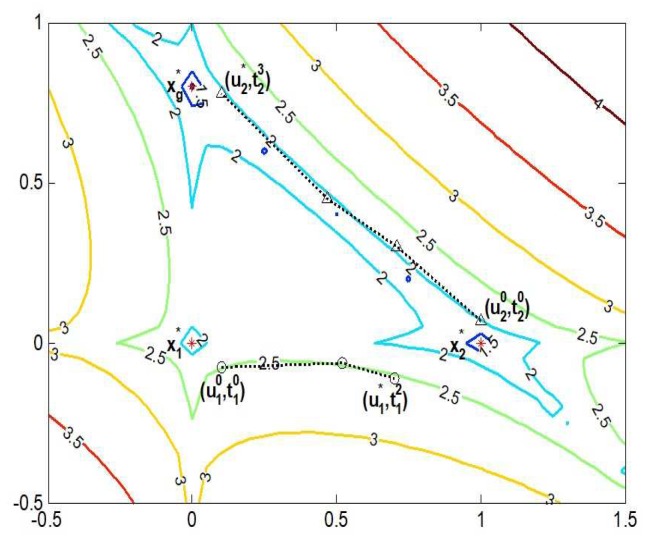
Global convergence process of the energy-level jumping algorithm in the contour map.

**Figure 5 sensors-19-02502-f005:**
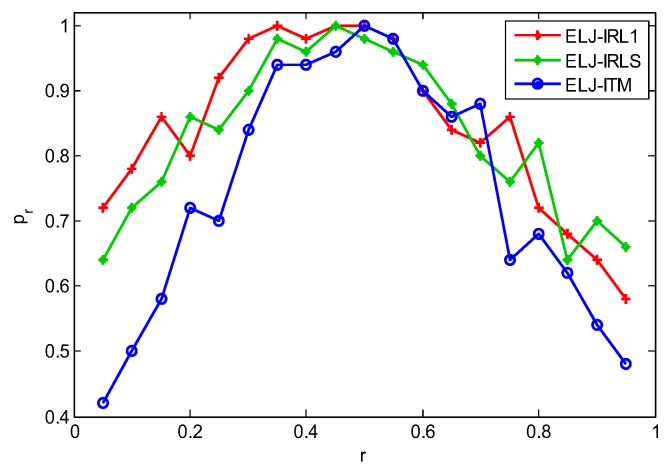
Probability of finding the globally optimal sparse solution for different rate *r*.

**Figure 6 sensors-19-02502-f006:**
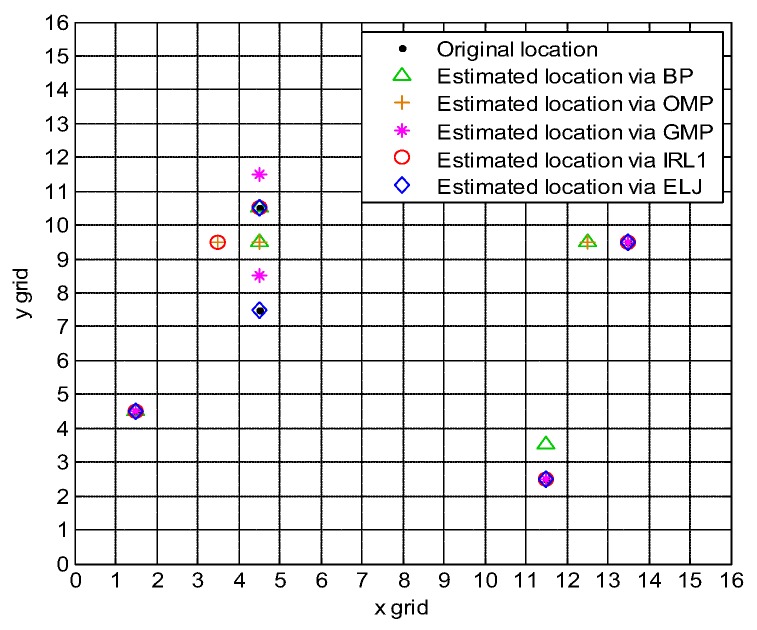
Comparison of target localization for five recovery algorithms.

**Figure 7 sensors-19-02502-f007:**
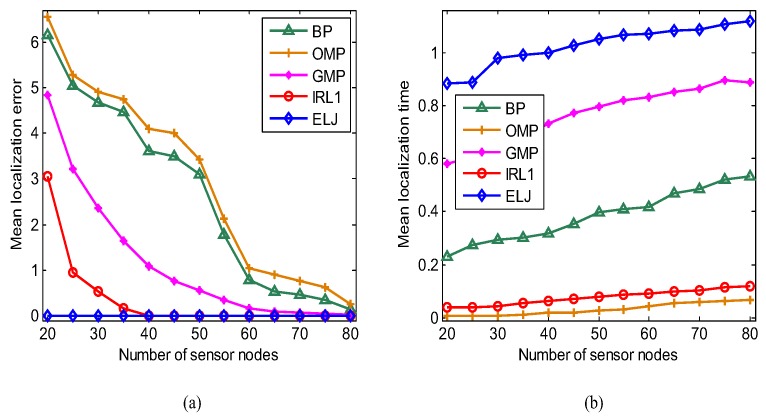
Performance analyses of five recovery algorithms for the different number of sensor nodes when K=5 and p=0.5. (**a**) Comparison of the mean localization error of BP, OMP, GMP, IRL1 and ELJ for the different number of sensor nodes. (**b**) Comparison of the mean localization time of BP, OMP, GMP, IRL1 and ELJ for the different number of sensor nodes.

**Figure 8 sensors-19-02502-f008:**
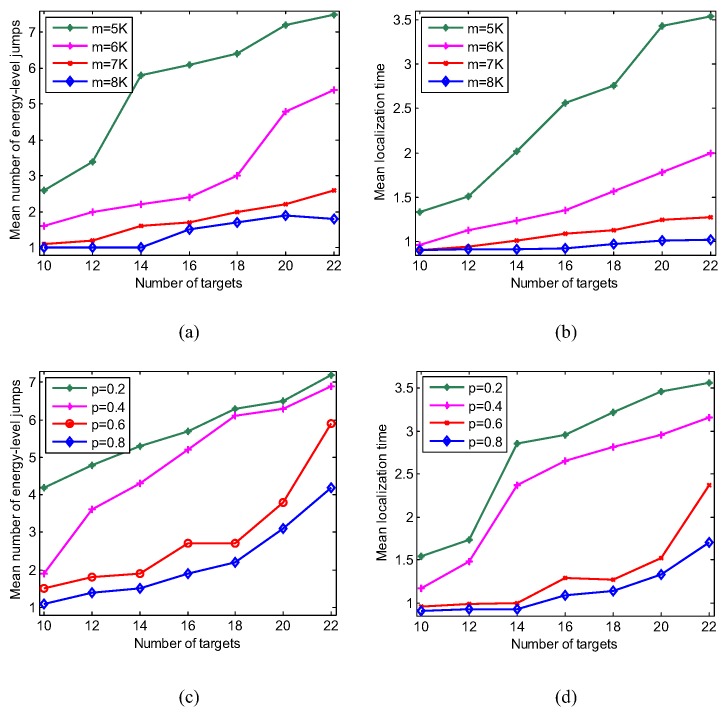
Performance analysis of ELJ for different K,m and *p*. (**a**) Comparison of the mean number of energy-level jumps for the different number of targets when p=0.5. (**b**) Comparison of the mean localization time for the different number of targets when p=0.5. (**c**) Comparison of the mean number of energy-level jumps for the different *p* when m=6K. (**d**) Comparison of the mean localization time for different *p* when m=6K.

**Figure 9 sensors-19-02502-f009:**
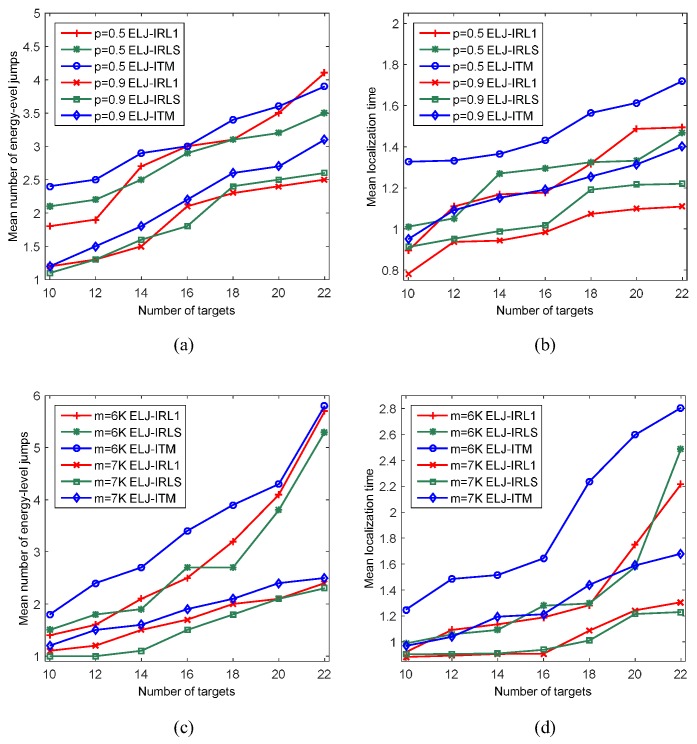
Performance analysis for ELJ using different local recovery algorithms. (**a**) Comparison of the mean number of energy-level jumps of ELJ-IRL1, ELJ-IRLS and ELJ-ITM for the different number of targets when m=7K. (**b**) Comparison of the mean localization time of ELJ-IRL1, ELJ-IRLS and ELJ-ITM for the different number of targets when m=7K. (**c**) Comparison of the mean number of energy-level jumps of ELJ-IRL1, ELJ-IRLS and ELJ-ITM for the different number of targets when p=0.7. (**d**) Comparison of the mean localization time of ELJ-IRL1, ELJ-IRLS and ELJ-ITM for the different number of targets when p=0.7.

**Figure 10 sensors-19-02502-f010:**
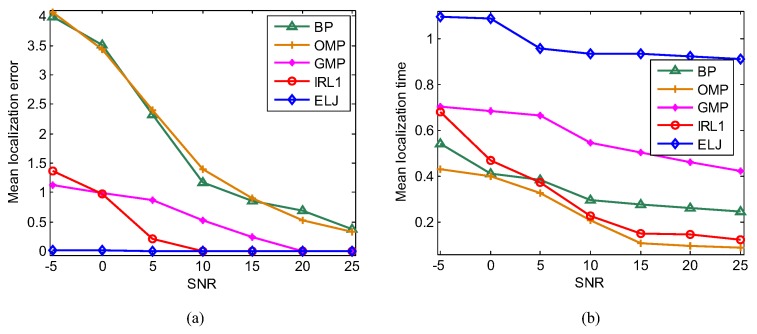
Performance analysis of five recovery algorithms for different measurement noise levels when K=10,m=65 and p=0.8. (**a**) Comparison of the mean localization errors of BP, OMP, GMP, IRL1 and ELJ for different SNR. (**b**) Comparison of the mean localization time of BP, OMP, GMP, IRL1 and ELJ for different SNR.

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
