# Peer review of "Energy-Level Jumping Algorithm for Global Optimization in Compressive Sensing-Based Target Localization"

_sensors, 2019, doi:10.3390/s19112502_

Round 1

Reviewer 1 Report

This is very nicely written paper, and the results are quite interesting. One notable challenge of the proposed algorithm, however, is its much larger time requirement than competing localization algorithms. While this is mentioned in the paper, it is not addressed in the detail it should be, given that time requirements are a significant drawback for such algorithms. The paper would be strengthened by discussing in what situations/conditions the ELJ is a reasonable algorithm choice, as well as future work that might reduce the time requirement for ELJ.

The paper is well organized and structured, but there are frequent grammatical errors that should be fixed to improve readability.

Author Response

We would like to   express our gratitude to the Reviewer for his (her) comments and suggestions,   which we carefully read and analysed. We provide below our point-by-point   reply and revisions in the paper to address the reviewer’ questions. For easy   reference, we first reproduce the reviewer’ comments in italic fonts.

Please note that all   modified parts in the revised version of the manuscript are marked with blue.

Point 1: One notable challenge of the proposed algorithm, however, is its much larger time requirement than competing localization algorithms. While this is mentioned in the paper, it is not addressed in the detail it should be, given that time requirements are a significant drawback for such algorithms. The paper would be strengthened by discussing in what situations/ conditions the ELJ is a reasonable algorithm choice, as well as future work that might reduce the time requirement for ELJ.

Response 1: Thanks for the reviewer’s comments. Solving the globally optimal solution of the objective optimization is a big challenge all the time. As we all know, many simply and convenient solving algorithms are easy to obtain suboptimal solutions. To find global minimizer, many solving algorithms with high complexity (e.g., genetic algorithm and swarm algorithm) have been provided. These algorithms spend time to search the globally optimal solution by testing a large amount of locally optimal solutions. Hence, finding the globally optimal solution is at the cost of high computation time.

In this paper, we also fact the above problem. We firstly try to propose the theoretical solving method for global optimization in the CS-based target localization. In the future work, we will focus on improve the performance of our ELJ algorithm by accelerating global convergence and reducing computation time.

In page 13: “Figure 6 shows that the localization via ELJ is very precise while the localization via BP, OMP, GMP and IRL1 has high estimation error. This is because  l_p-norm minimization can provide sparser solution than that of l_0-norm and l_1-norm minimization, and ELJ uses the global minimizer to obtain the accurately estimated locations of targets while BP, OMP, GMP and IRL1 use the local minimizer to locate targets leading to the missed targets and the false targets. Furthermore, the localization time of BP, OMP, GMP, IRL1 and ELJ is 0.4584s, 0.0487s, 0.7874s, 0.0808s and 1.1099s, respectively. Although the localization time of four algorithms is less than that of ELJ, they provide incorrectly estimated locations of some of targets. It is observed that the performance of the ELJ algorithm is a trade-off between localization accuracy and localization time. If the primary aim of CS-based target localization is to determine the locations of targets with high precision, we need to tolerant the slight increasing of computation time of ELJ compared to that of BP, OMP, GMP and IRL1. On the other hand, satisfying the real-time requirement of CS-based target localization is at the cost of localization accuracy of ELJ, namely we only find a better locally optimal sparse solution rather than the globally optimal sparse solution to locate targets in limited localization time.

In page 17:

ž To reduce localization delay, the ELJ algorithm will be improved by accelerating global convergence and reducing computation time.

Point 2: The paper is well organized and structured, but there are frequent grammatical errors that should be fixed to improve readability.

Response 2: Thanks for the reviewer’s comments. We have checked carefully for the writing errors. The modified parts in the revised version of the manuscript are marked with blue.

In page 2: “A sparse solution modelling the locations of targets was then reconstructed by the basis pursuit (BP) algorithm [13] with high computational complexity.”

In page 2: “Another greedy matching pursuit (GMP) algorithm [15] tried to select an optimal 1-sparse vector at each iteration (i.e., the location of a target), which inevitably led to much localization time.”

In page 3: “To maintain the signature map, the fingerprinting-based method has to exchange a large number of data among sensors.”

In page 4: “As we know many physical quantities are intrinsically sparse, so they can be represented by few nonzero expansion coefficients with respect to a suitable expansion basis [40]-[43].”

In page 6: “In order to better illustrate the relationship between the concept of energy level and our sparse recovery algorithm, we use an example to display the process of energy-level jumping for solving global optimization.”

In page 8: “Now, problem (13) is transformed into the following initial value problem of differential equations to establish a homotopy curve u=u(t)”

In page 9: “then the limit of this sequence  u_l^*will be in an attraction basin with the lower energy level.”

In page 10: “Algorithm 1 displays that the iterative process is stopped by comparing the energy values of two locally optimal sparse solutions”

In page 12: “Applying the frequently used IRL1, IRLS and ITM to search a local minimizer, Figure 5 shows that ELJ has the high probability to obtain the global minimizer”

In page 14: “Furthermore, the localization time of BP, OMP, GMP, IRL1 and ELJ is 0.4584s, 0.0487s, 0.7874s, 0.0808s and 1.1099s, respectively.”

In page 15: “Here, the experimental results reveal that our ELJ algorithm can obtain the better localization performance when using IRL1 and IRLS to solve local optimization.”

In page 17: “Our simulation results show that the ELJ algorithm can provide the more accurately estimated locations of targets compared to other existing sparse recovery algorithms.”

Reviewer 2 Report

Authors propose an energy-level jumping algorithm to address the CS-based target problem. The concerns are the following: 

- The state of the art is not adequate. Please include recent (hybrid) works that combine RSS measurements with ToA, AoA,.. 

- The model in (5) is too optimistic. Authors assume that the bias term is 0. That is, line of sight is assumed. Can this method be generalized to the more realistic scenario where the bias term is not 0 (i.e. to a non line of sight scenario). If the algorithm is only applicable to LOS please state it clearly. If it can be generalized to the NLOS, explain why. 

-The algorithm is a centralized one. Can it be generalized to a distributed setup? This is a pertinent issue. Please discuss. If the algorithm can only be used in a centralized setup please state it clearly. If it can be generalized to a distributed scenario, explain why. 

-Theorem 5.4 is not really a theorem. That is, it is rather strange to use a word mainly in a theorem. Please reformulate.

-  The convergence of the method depends on the choice of r (Sec 6.1.). In practice,  how to choose/estimate r? In an offline phase? The choice of r depends on what parameters? Please discuss.

- From Fig. 7 one can conclude that the new method is more time consuming,  i.e. it is order of 10 times more time consuming when compared to IRL1. The performance gain justifies this increase? 

- In Fig.7a the performance of the new method is independent of the number of sensors. This is rather strange. What is the explanation? Please discuss.

-From Sec. 6.4. we can conclude that the performance of the new method depends on the choice of p. . In practice,  how to choose/estimate p? In an offline phase? The choice of r depends on what parameters? Please discuss.

- Real measurements are missing. It is rather cheap to install a WSN and perform measurements. From this reviewer's experience, matlab results and results extracted from real measurements can be rather different. That is, the conclusions can be rather different, e.g. a method that performs well in matlab can have poor behavior with real measurements. 

- Authors use known tools (CS and energy-level jumping) to address a known problem. So the overall contribution is modest.

Author Response

We would like to   express our gratitude to the Reviewer for his (her) comments and suggestions,   which we carefully read and analysed. We provide below our point-by-point   reply and revisions in the paper to address the reviewer’ questions. For easy   reference, we first reproduce the reviewer’ comments in italic fonts.

Please note that all   modified parts in the revised version of the manuscript are marked with blue.

Point 1: The state of the art is not adequate. Please include recent (hybrid) works that combine RSS measurements with ToA, AoA,.. 

Response 1: Thanks for the reviewer’s comments. We have added the recent (hybrid) works that combine RSS measurements with ToA, AoA.

In page 3:The hybrid schemes based on angle of arrival-time of arrival (AoA-ToA), time of arrival-received signal strength (ToA-RSS), and angle of arrival-received signal strength (AoA-RSS) signals are proposed to further enhance the localization performance [23][24].

[23] Gante, A.D.; Siller, M. A survey of hybrid schemes for location estimation in wireless sensor networks. Procedia Technology, 2013, 7, 377-383.

[24] MW, K.; AH, K.; Salman, N. Optimized hybrid localisation with cooperation in wireless sensor networks. IET Signal Processing, 2017, 11, 341-348.

Point 2: The model in (5) is too optimistic. Authors assume that the bias term is 0. That is, line of sight is assumed. Can this method be generalized to the more realistic scenario where the bias term is not 0 (i.e. to a non line of sight scenario). If the algorithm is only applicable to LOS please state it clearly. If it can be generalized to the NLOS, explain why.   

Response 2: Thanks for the reviewer’s comments.  We did not explain clearly the measurement noise in the path loss model (5). We have added the explanation of the measurement noise due to the reviewer’s comment.

In the last experiment, the measured deviation in the path loss model and the measured deviation of compressive sampling are put together as the measurement noise. The experimental results show that our proposed algorithm has the better performance of noise reduction than some conventional recovery algorithms. Therefore, our proposed algorithm can be generalized to the NLOS.

In page 5: “According to the path loss model [52], the sensor node i senses the RSS from the target in the grid j, which follows                               

                                                  p_ij=p_0-10ηlog(d_ij/d_0)                                                                 (5)

where  p_ij is the received signal power,  d_ij is the Euclidean distance,  d_0 is a reference distance with the received power p_0,  η is the path loss coefficient with typically values between 2 and 4. If the RSS is affected by multi-path fading and shadowing effect in practice, some signal pre-processing methods including averaging multiple measurements and Kalman filtering can be applied to reduce the effect of measurement noise.

In page 15: “The complex and volatile environment leads to the measurement noise including the measured deviation in the path loss model and the measured deviation of compressive sampling, which is set as Gaussian white noise in our experiments.”

Point 3: The algorithm is a centralized one. Can it be generalized to a distributed setup? This is a pertinent issue. Please discuss. If the algorithm can only be used in a centralized setup please state it clearly. If it can be generalized to a distributed scenario, explain why. 

Response 3: Thanks for the reviewer’s comments. We ignored to discuss the application scenario of our ELJ algorithm. We have added the discussion due to the reviewer’s comment.

The algorithm can be generalized to a distributed setup. A sensor node can locate adjacent targets by applying our ELJ algorithm after it receives the sufficient number of measurements from its neighbour nodes. On the other hand, a clusterhead can also locate targets in its cluster by applying our ELJ algorithm after it receives the sufficient number of measurements from its member nodes. Therefore, our proposed algorithm can be applied to the centralized setup and distributed setup. We have added the explanation for the application scenario of our ELJ algorithm.

In page 10: “This advantage is beneficial to apply our ELJ algorithm to some practical localization scenarios in centralized setup, which have no prior knowledge of the number of targets. On the other hand, our ELJ algorithm can also be applied to a distributed scenario. For example, a clusterhead can locate targets in its cluster by applying the ELJ algorithm after it receives the sufficient number of measurements from its member nodes.

Point 4: Theorem 5.4 is not really a theorem. That is, it is rather strange to use a word mainly in a theorem. Please reformulate.

Response 4: Thanks for the reviewer’s comments. The presentation for evaluating the convergence efficiency of ELJ is not precise. We have reformulated the presentation due to the reviewer’s comment.

In page 12: “If applying different reconstruction algorithms (e.g., IRL1, IRLS and ITM) to find a local minimizer of problem (7), the number of energy-level jumps can be used to evaluate the convergence efficiency of ELJ in terms of the following analysis. According to the convergence theorem 5.3, the solving process of ELJ is divided into two stages. In first stage, a recovery algorithm is applied to find a locally optimal sparse solution, which is limited by the initial point x_0. In second stage, no matter what kinds of recovery algorithms we use, the same homotopy method is used to make the local solution jump from the current attraction basin to another one. This means that our homotopy method has little impact on the number of energy-level jumps. Therefore, the number of energy-level jumps of our ELJ algorithm mainly depends on the initial point  x_0 and the used recovery algorithm.”

Point 5: The convergence of the method depends on the choice of r (Sec 6.1.). In practice, how to choose/estimate r? In an offline phase? The choice of r depends on what parameters? Please discuss.

Response 5: Thanks for the reviewer’s comments. We have added the discussion due to the reviewer’s comment.

According to the theoretical analysis in Figure 3, we stimulate the locally optimal solution x_l^* to the excited state by absorbing the energy ε_l, which is proportional to the energy of  x_l^*. We prefer to choose the proportional coefficient  r from [0.3, 0.5] due to the experimental results in Figure 5. In practice,  r can be set any value from 0.3 to 0.5, because the connected contour lines of the objective function between two attraction basins are infinite in Figure 3.

In page 13: “To make ELJ become a descent algorithm in terms of Theorem 5.3, we prefer to choose r from [0.3, 0.5]. In fact, any value from 0.3 to 0.5 can be assigned to r without affecting the process of energy jumping, because the connected contour lines of the objective function between two attraction basins are infinite in Figure 3.

Point 6: From Fig. 7 one can conclude that the new method is more time consuming, i.e. it is order of 10 times more time consuming when compared to IRL1. The performance gain justifies this increase? 

Response 6: Thanks for the reviewer’s comments. Solving the globally optimal solution of the objective optimization is a big challenge all the time. As we all know, many simply and convenient solving algorithms are easy to obtain suboptimal solutions. To find global minimizer, many solving algorithms with high complexity (e.g., genetic algorithm and swarm algorithm) have been provided. These algorithms spend time to search the globally optimal solution by testing a large amount of locally optimal solutions. Hence, finding the globally optimal solution is at the cost of high computation time.

   In this paper, we also fact the above problem. We firstly try to propose the theoretical solving method for global optimization in the CS-based target localization. In the future work, we will focus on improve the performance of our ELJ algorithm by accelerating global convergence and reducing computation time.

In page 13: “Figure 6 shows that the localization via ELJ is very precise while the localization via BP, OMP, GMP and IRL1 has high estimation error. This is because  l_p-norm minimization can provide sparser solution than that of l_0-norm and l_1-norm minimization, and ELJ uses the global minimizer to obtain the accurately estimated locations of targets while BP, OMP, GMP and IRL1 use the local minimizer to locate targets leading to the missed targets and the false targets. Furthermore, the localization time of BP, OMP, GMP, IRL1 and ELJ is 0.4584s, 0.0487s, 0.7874s, 0.0808s and 1.1099s, respectively. Although the localization time of four algorithms is less than that of ELJ, they provide incorrectly estimated locations of some of targets. It is observed that the performance of the ELJ algorithm is a trade-off between localization accuracy and localization time. If the primary aim of CS-based target localization is to determine the locations of targets with high precision, we need to tolerant the slight increasing of computation time of ELJ compared to that of BP, OMP, GMP and IRL1. On the other hand, satisfying the real-time requirement of CS-based target localization is at the cost of localization accuracy of ELJ, namely we only find a better locally optimal sparse solution rather than the globally optimal sparse solution to locate targets in limited localization time.

In page 17:

ž To reduce localization delay, the ELJ algorithm will be improved by accelerating global convergence and reducing computation time.

Point 7: In Fig.7a the performance of the new method is independent of the number of sensors. This is rather strange. What is the explanation? Please discuss.

Response 7: Thanks for the reviewer’s comments.  The performance of our proposed algorithm is also dependent of the number of sensors. In Fig.7(a), the change of localization errors of our proposed method is not obvious, because the change amplitudes of localization errors of the other algorithms are very lager. We have added the illustration.

In page 14: “As can be seen, five targets are accurately located as the number of measurements is more than 25 (i.e., m≥5K), so the target localization via ELJ achieves the best performance as well as greatly reduces the sensing consumption of sensors.”

Point 8: From Sec. 6.4. we can conclude that the performance of the new method depends on the choice of p. In practice, how to choose/estimate p? In an offline phase? The choice of p depends on what parameters? Please discuss.

Response 8: Thanks for the reviewer’s comments. As reviewer say, the performance of our ELJ algorithm depends on the choice of p. In general, the performance of many well-known sparse recovery algorithms for l_p-norm minimization all depends on the choice of p, because the value of  p affects the non-convex property of   l_p -norm minimization. The sparse recovery performance can be improved when the value of  p becomes larger. In Fig. 8(c), the experimental results shows that we prefer to choose p from [0.6,1) to reduce the mean energy-level jumps, i.e., the computation cost. In most cases, the researcher will pre-set a larger value of p before performing the sparse recovery algorithm.

In page 15:In practice, we can pre-set a larger value of p to accelerate the global convergence of the ELJ algorithm.

Point 9: Real measurements are missing. It is rather cheap to install a WSN and perform measurements. From this reviewer's experience, matlab results and results extracted from real measurements can be rather different. That is, the conclusions can be rather different, e.g. a method that performs well in matlab can have poor behavior with real measurements. 

Response 9: Thanks for the reviewer’s comments. In this paper, we try to propose target localization via  l_p -norm minimization in theory, and test the performance of our proposed algorithm by the simulation in Matlab. Therefore, the practical scenarios are not considered. In the future work, we will focus on the install a WSN and perform target localization to test the real performance of our proposed algorithm.

In page 17:

ž Various types of propagation models replacing the path loss model will be applied to test the real localization performance of the proposed algorithm.

ž Installing a WSN in a smart community, various types of application scenarios (e.g., CS-based target localization for people and car) will be considered to test the practicability of the proposed algorithm such that the parameters are more reasonably set.

Point 10: Authors use known tools (CS and energy-level jumping) to address a known problem. So the overall contribution is modest.

Response 10: Thanks for the reviewer’s comments. Solving the globally optimal solution of the non-convex optimization problem is a big challenge all the time.  l_p -norm minimization in CS-based target localization is a representative non-convex optimization problem. An effective and universal solving method for the non-convex optimization problem can be motivated by designing the recovery algorithm for l_p -norm minimization. The conventional recovery algorithms are easy to obtain suboptimal solutions, because the iterative solution cannot jump out of the attraction basin of a local minimizer. To the best of our knowledge, this is the first time to make the iterative solution jump from one attraction basin to another one by constructing a connected channel between two attraction basins. As we can see, this method very effectively solves global optimization after achieving a local minimizer. Therefore, our proposed algorithm is not restricted to  l_p -norm minimization in CS-based target localization, and it can be applied to more non-convex optimization problems. Our major contribution in this paper is to provide the new idea that finds the globally optimal solution by constructing connected channels among the different attraction basins. To the best of our knowledge, this idea has not been mentioned in the past research, especially CS-based practical problems.

Reviewer 3 Report

In this paper, the RSS-fingerprint based target localization is particularly investigated.

Although the massive signature map of RSS-fingerprints can guarantee the accuracy of target localization, the high computation complexity and storage requirement of the sensors conflict the purpose of using WSN, i.e. low cost and low complexity.

From the viewpoint of complexity reduction, conducting the target localization with significantly reduced RSS-fingerprint data from sensors is formulated as a sparse recovery problem and using compressive sensing to resolve.

Energy-jumping algorithm is utilized in resolving local optimal issue in compressive sensing and attaining the global optimal solution.

Integrating Energy-jumping algorithm into compressive sensing is an interesting idea and innovative. The authors well organize the paper and clearly present the workflow of solving the problem.

The accuracy of locating the multiple targets is higher than those of the counterparts used as benchmarks. However, the time of convergence of the proposed algorithm seems to be the weakest part. It makes the proposed algorithm less useful, however, still has significant values to enrich the capacity of related research.

I suggest that the authors may revise and add the results which show an improvement of convergence speed of the proposed algorithm, even it means it requires some penalty, such as complexity, more RSS-signatures, etc. And properly address the trade-off and balance point of the complexity, convergence speed and accuracy of the proposed algorithm.

and there are some typos: "reconvery" algorithm in page 6.

Author Response

We would like to   express our gratitude to the Reviewer for his (her) comments and suggestions,   which we carefully read and analysed. We provide below our point-by-point   reply and revisions in the paper to address the reviewer’ questions. For easy   reference, we first reproduce the reviewer’ comments in italic fonts.

Please note that all   modified parts in the revised version of the manuscript are marked with blue.

Point 1: The accuracy of locating the multiple targets is higher than those of the counterparts used as benchmarks. However, the time of convergence of the proposed algorithm seems to be the weakest part. It makes the proposed algorithm less useful, however, still has significant values to enrich the capacity of related research.

Response 1: Thanks for the reviewer’s comments. Solving the globally optimal solution of the objective optimization is a big challenge all the time. As we all know, many simply and convenient solving algorithms are easy to obtain suboptimal solutions. To find global minimizer, many solving algorithms with high complexity (e.g., genetic algorithm and swarm algorithm) have been provided. These algorithms spend time to search the globally optimal solution by testing a large amount of locally optimal solutions. Hence, finding the globally optimal solution is at the cost of high computation time.

     In this paper, we also fact the above problem. We firstly try to propose the theoretical solving method for global optimization in the CS-based target localization. In the future work, we will focus on improve the performance of our ELJ algorithm by accelerating global convergence and reducing computation time.

In page 17:

ž To reduce localization delay, the ELJ algorithm will be improved by accelerating global convergence and reducing computation time.

Point 2: I suggest that the authors may revise and add the results which show an improvement of convergence speed of the proposed algorithm, even it means it requires some penalty, such as complexity, more RSS-signatures, etc. And properly address the trade-off and balance point of the complexity, convergence speed and accuracy of the proposed algorithm.

Response 2: Thanks for the reviewer’s comments. We ignored to discuss the trade-off and balance point of the complexity, convergence speed and accuracy of the proposed algorithm. We have added the corresponding discussion.

In page 13: “Figure 6 shows that the localization via ELJ is very precise while the localization via BP, OMP, GMP and IRL1 has high estimation error. This is because l_p-norm minimization can provide sparser solution than that of  l_0-norm and l_1-norm minimization, and ELJ uses the global minimizer to obtain the accurately estimated locations of targets while BP, OMP, GMP and IRL1 use the local minimizer to locate targets leading to the missed targets and the false targets. Furthermore, the localization time of BP, OMP, GMP, IRL1 and ELJ is 0.4584s, 0.0487s, 0.7874s, 0.0808s and 1.1099s, respectively. Although the localization time of four algorithms is less than that of ELJ, they provide incorrectly estimated locations of some of targets. It is observed that the performance of the ELJ algorithm is a trade-off between localization accuracy and localization time. If the primary aim of CS-based target localization is to determine the locations of targets with high precision, we need to tolerant the slight increasing of computation time of ELJ compared to that of BP, OMP, GMP and IRL1. On the other hand, satisfying the real-time requirement of CS-based target localization is at the cost of localization accuracy of ELJ, namely we only find a better locally optimal sparse solution rather than the globally optimal sparse solution to locate targets in limited localization time.

Point 3: There are some typos: "reconvery" algorithm in page 6.

Response 3: Thanks for the reviewer’s comments. We have checked carefully for the typos. The modified parts in the revised version of the manuscript are marked with blue.

In page 2: “A sparse solution modelling the locations of targets was then reconstructed by the basis pursuit (BP) algorithm [13] with high computational complexity.”

In page 2: “Another greedy matching pursuit (GMP) algorithm [15] tried to select an optimal 1-sparse vector at each iteration (i.e., the location of a target), which inevitably led to much localization time.”

In page 3: “To maintain the signature map, the fingerprinting-based method has to exchange a large number of data among sensors.”

In page 4: “As we know many physical quantities are intrinsically sparse, so they can be represented by few nonzero expansion coefficients with respect to a suitable expansion basis [40]-[43].”

In page 6: “In order to better illustrate the relationship between the concept of energy level and our sparse recovery algorithm, we use an example to display the process of energy-level jumping for solving global optimization.”

In page 8: “Now, problem (13) is transformed into the following initial value problem of differential equations to establish a homotopy curve u=u(t)”

In page 9: “then the limit of this sequence  u_l^* will be in an attraction basin with the lower energy level.”

In page 10: “Algorithm 1 displays that the iterative process is stopped by comparing the energy values of two locally optimal sparse solutions”

In page 12: “Applying the frequently used IRL1, IRLS and ITM to search a local minimizer, Figure 5 shows that ELJ has the high probability to obtain the global minimizer”

In page 14: “Furthermore, the localization time of BP, OMP, GMP, IRL1 and ELJ is 0.4584s, 0.0487s, 0.7874s, 0.0808s and 1.1099s, respectively.”

In page 15: “Here, the experimental results reveal that our ELJ algorithm can obtain the better localization performance when using IRL1 and IRLS to solve local optimization.”

In page 17: “Our simulation results show that the ELJ algorithm can provide the more accurately estimated locations of targets compared to other existing sparse recovery algorithms.”

Round 2

Reviewer 2 Report

Include more references  on hybrid techniques  (RSS+AoA, RSS+ToA, ..) as it is the state-of.the art.

Author Response

  We would like to express our gratitude to the Reviewer for his (her) comments and suggestions, which we carefully read and analysed. We provide below our point-by-point reply and revisions in the paper to address the reviewer’ questions. For easy reference, we first reproduce the reviewer’ comments in italic fonts.    Please note that all modified parts in the revised version of the manuscript are marked with blue.    

Point 1: Include more references on hybrid techniques (RSS+AoA, RSS+ToA, ..) as it is the state-of.the art.  

Response 1: Thanks for the reviewer’s comments. We have added more references to introduce the hybrid techniques.

In page 3: “Some hybrid schemes based on angle of arrival-time of arrival (AoA-ToA), time of arrival-received signal strength (ToA-RSS), and angle of arrival-received signal strength (AoA-RSS) signals are proposed to further enhance the localization performance[23][24]. To improve the location accuracy of hybrid (AoA-ToA) localization systems, a linear least squares (LLS) algorithm was used to obtain the location coordinates. Furthermore, a weighted linear least squares (WLLS) algorithm exhibited better localization performance than the LLS algorithm by utilizing the information present in the covariance of the incoming signals [25][26]. The authors of [27] proved that hybridization of different types of measurements (TOA/RSS) could enhance localization accuracy from Cramer-Rao lower bound (CRLB) analysis, but signal-to-noise ratio, path-loss exponent, as well as anchors placement all affected the hybrid TOA/RSS LLS localization accuracy in different ways. The authors of [28] applied a RSS/AOA hybrid positioning algorithm to increase the accuracy of localization when searching and rescuing the survivals in huge disaster. ”

[25] Khan, M.W.; Salman, N.; Kemp, A.H. Enhanced hybrid positioning in wireless networks I: AoA-ToA. In: International Conference on Telecommunications and Multimedia 2014; pp. 86-91.

[26] Khan, M.W.; Salman, N.; Kemp, A.H. Enhanced hybrid positioning in wireless networks II: AoA-ToA. In: International Conference on Telecommunications and Multimedia 2014; pp. 92-97.

[27] Zhu, X.X.; Wang, Y.; Guo, Y.L.; Chen, J.Y.; Li N.; Zhang, B. Effect of inaccurate range measurements on hybrid TOA/RSS linear least squares localization. In: Proceedings of the 2015 International Conference on Communications, Signal Processing, and Systems 2016, 386; pp. 523-530.

[28] Tang, S.Y.; Shu, X.M.; Hu, J.; Zhou, R.; Shen S.F.; Cao, S.Y. Study on RSS/AOA hybrid localization in life detection in huge disaster situation. Natural Hazards 2019, 95, 569-583.